



# Acceptance of wind energy – Theoretical concepts, empirical drivers and some open questions

Michael Ruddat[1]

[1]University of Stuttgart, ZIRIUS – Research Center for Interdisciplinary Risk and Innovation Studies, Seidenstr. 36, 70174 Stuttgart GERMANY

*Correspondence to*: Michael Ruddat (michael.ruddat@zirius.uni-stuttgart.de)

**Abstract.** The further development of wind energy is of major importance for the success of the energy system transformation in Germany and elsewhere. This transition process is not an easy task. For example, the yearly installed capacity of wind energy onshore in Germany is declining since 2017. Only relatively few new wind turbines were constructed especially in 2019. Problems are for example minimum distance requirements (e. g. residential areas, air safety), the high complexity of planning processes and local protests. Social science research has now dealt with the topic of wind energy acceptance for quite some time. On the one hand, the specific kind of acceptance (e.g. local acceptance) has been subject to scientific discourse. On the other hand, different empirical drivers (e. g. perceived distributional or procedural fairness, trust in relevant actors of the transformation process, risk/benefit perceptions, participation) have been of special interest. This review deals with central theoretical concepts as well as qualitative and quantitative empirical findings of social science research concerning the acceptance of wind energy in Germany and elsewhere. Although there has been already a lot of valuable scientific work done, there are still some open questions left.

**Keywords:** Acceptance, wind energy, fairness, trust, risk/benefit perceptions, participation

## 1        Introduction

The further development of wind energy is of major importance for the success of the energy system transformation in Germany and elsewhere. This transition process is not an easy task. The yearly installed capacity of wind energy onshore in Germany is declining since 2017. Only relatively few new wind turbines were constructed especially in 2019. Problems are for example minimum distance requirements (e. g. residential areas, air safety), the high complexity of planning processes and local protests (AEE, 2020; Di Nucci et al., 2020). In Great Britain, a change in the planning law to allow for local control together with the removal of financial support for onshore wind energy led to a dramatic decline since 2015 (Harper et al., 2019). In 2001, the biggest wind energy project in the Netherlands failed because the government refused to negotiate with environmental groups (Wolsink, 2007b). According to Hoen and colleagues, wind parks in the USA are moving closer to residential areas raising the risk of conflict (Hoen et al., 2019). All in all, acceptance of wind energy projects seems to be of crucial importance for the success of the energy system transformation.



Social science research has now dealt with the topic of wind energy acceptance for quite some time (see for example Aitken, 2010a; Breukers and Wolsink, 2007; Devine-Wright, 2007; Hoen et al., 2019; Jones and Eiser, 2010; Krohn and Damborg, 1999; Pasqualetti, 2001; Sonnberger and Ruddat, 2017). On the one hand, the specific kind of acceptance (e. g. socio-political acceptance, market acceptance or community acceptance, Wüstenhagen et al., 2007) has been subject to scientific

discourse. On the other hand, different empirical drivers (e. g. perceived distributional or procedural fairness, trust in relevant actors of the transformation process, risk/benefit perceptions, participation) have been of special interest. This review[i] deals with central theoretical concepts as well as qualitative and quantitative empirical findings of social science research concerning the acceptance of wind energy in Germany and elsewhere. Although there has been already a lot of valuable scientific work done, there are still some open questions left.

**2        Theoretical concepts of acceptance and research results**

The iridescent concept of (risk) acceptance has already been defined and conceptualized by many scholars in social sciences (e.g. Cohen et al., 2014; Hübner and Hahn, 2013; Renn, 2008; Schweizer-Ries, 2008; Sauter and Watson, 2007; Upham et al., 2015; Wüstenhagen et al., 2007). From an analytical point of view, it can be understood as the balancing of pros and cons (benefit and risks) with a final decision to support or oppose an action, project or technology (Ajzen and Fishbein, 1980;

Ajzen, 1991; Fischhoff et al., 1981; Groth and Vogt, 2014; Renn, 1984). This process can be related to positive or negative attitudes[ii] (Cohen et al., 2014, Schäfer und Keppler, 2013) as well as behavioural intentions and behaviour itself (Petermann and Scherz, 2005).

Another thought of school emphasizes the importance of emotions, feelings and moods for the judgement of the acceptability of risks (Böhm and Tanner, 2019; Klinke and Renn, 2002; Schwarz, 2002; Slovic et al., 2004). One example is the affect

heuristic implying a reversed relationship between emotions and risk/benefit perceptions: Positive emotions correlate with high benefit and low risk whereas negative emotions are associated with low benefit and high risk (Böhm and Tanner, 2019). Accordingly, positive emotions can be connected to positive attitudes and vice versa.

The close connection between acceptance and attitudes is demonstrated by Upham and colleagues who define acceptance as "[...] a favourable or positive response (including attitude, intention, behaviour and – where appropriate – use) relating to a

proposed or in situ technology or socio-technical system, by members of a given social unit (country or region, community or town and household, organization)" (Upham et al., 2015: 103). Schweizer-Ries and colleagues (e. g. Schweizer-Ries, 2008; Hildebrand et al., 2017) also make the link between attitude and action[iii] and come up with a rather complex acceptance typology including supporters (positive attitude / intention or action), advocates (positive attitude / no intention or action), opponents (negative attitude / no intention or action) and protesters (negative attitude / intention or action).

The definition of Upham et. al can be analytically separated in *acceptance subject* ("members of a given social unit"), *acceptance object* ("proposed or in situ technology or socio-technical system") and *acceptance context* ("country or region, community or town and household, organization")[iv]. The three elements of subject, object and context are an additional



concept to analyse acceptance (Hüsing et al., 2002; Lucke, 1995; Schäfer and Keppler, 2013). Another prominent typology with reference to the acceptance object is the one of Wüstenhagen and colleagues. They differentiate between socio-political

acceptance, market acceptance and community acceptance (sometimes also called "local acceptance"). *Socio-political acceptance* is located on the level of society and refers to technologies like wind energy and political programs promoting these technologies. Acceptance subjects are the public, central stakeholders and politicians. *Market acceptance* means purchasing and use of goods and services in free markets. In this case, the acceptance objects are products like roof-top solar collectors or services like green energy. Acceptance subjects include consumers, investors and companies. *Community*

*acceptance* refers to technological projects like wind farms in the vicinity of residential areas. Acceptance subjects are local residents, stakeholders and authorities, respectively (Wüstenhagen et al., 2007).

Despite the varying concepts and measurements, results are surprisingly often similar. For example, the acceptance typology of Schweizer-Ries and colleagues has been empirically applied on basis of data coming from a representative telephone survey of the German population in 2015 (n=2.006). The survey focused on the perception and evaluation of renewables,

especially the local acceptance. This local acceptance of wind parks, solar farms and high-tension power lines was operationalized by a 500 m distance to the respondent's place of residence. Results show high acceptance rates for solar farms as well as a considerable potential for protest and opposition against wind farms and high-tension power lines (Ruddat and Sonnberger, 2019). This finding was successfully replicated by Liebe and Dobers. They use a non-representative sample from a German panel to conduct an online survey in 2013 (n=3.192). The two researchers analyse acceptance of and

potential protest against the construction of diverse renewable power plants (wind/solar/biomass) and a natural gas-fired plant within a 10-kilometer radius of the respondent's place of residence. Irrespective of the different distances, survey methodology and operationalisation, the acceptance ranking for solar energy (first place) and wind energy (second place) was the same as reported by Ruddat and Sonnberger. The third place goes to biomass energy followed by natural gas on fourth place. The ranking for potential protest is just reversed (Liebe and Dobers, 2019).

While public survey research often reports high general support for wind energy in different countries (e. g. Devine-Wright, 2005; 2007; FA Wind, 2020; Krohn and Damborg, 1999; Steentjes et al., 2017), there are low success rates in the implementation of it (e. g. AEE, 2020; Harper et al., 2019). This finding is called the "social gap" (Bell et al., 2005)[v]. There is also a remarkable difference between general and local support. Data from the Special Eurobarometer 364 show an all in all high socio-political acceptance of wind energy in twelve different European countries (n=13.091, European Commission,

2011: 161). The same is all in all true for acceptance of offshore and onshore wind energy, respectively, as can be seen in data from a representative cross-national survey in Germany, France, Norway and the UK. Only solar energy is rated better and there is a clear preference for wind energy in comparison to oil, coal and nuclear power (n=4.048, Steentjes et al., 2017: 27). In contrast to this, local acceptance is more problematic. For example, researchers in Germany asked for the acceptance of solar farms, wind farms and high-tension power lines in 500 m distance to the respondents' home (representative

telephone survey, n=2.006). While more than half of the respondents would have no problems with solar parks in their



neighbourhood, only 35% would be willing to accept a wind park in their vicinity[vi]. High-tension power lines are perceived as even more negative (Sonnberger and Ruddat, 2016: 36).

Altogether, acceptance means a positive evaluation of a topic (like wind energy or wind parks) by a social group (e. g. stakeholders, residents, the public) under certain circumstances (e. g. cultural or institutional context) that can have consequences for individual behaviour. Correspondingly, non-acceptance means a negative evaluation of a topic by a social group under certain circumstances that can have consequences for individual behaviour. This definition can be the basis for the comparison of (quantitative) studies. It is also useful for the investigation of possible empirical drivers of acceptance.

## 3      Empirical drivers of wind energy acceptance

Social science research has identified several empirical drivers of wind energy acceptance. This review concentrates on prominent quantitative as well as qualitative studies from for example Australia, Great Britain, Germany, the Netherlands, Scotland, Sweden, Switzerland and the USA in the last centuries. Although it is a relevant sample, it does of course not claim to be complete.

### 3.1      Visual effects and place attachment

Wind turbines as well as wind farms can be designed in different ways. For example, Pasqualetti compares the differences between wind parks in the USA and Europe with respect to colour, uniformity of heights etc. He concludes that the European design with an all in all better compatibility with existing landscape seems to be far more acceptable for residents (Pasqualetti, 2001). This means that wind turbines can have negative as well as positive visual effects (Hoen et al., 2019, Krohn and Damborg, 1999). They are not generally being perceived as ugly and can also symbolize progress (survey data from UK, n=1.286, Devine-Wright, 2005: 128f, see also Swofford and Slattery, 2010; van der Horst and Toke, 2010). But irrespective of how well they are designed, they are not invisible (Pasqualetti, 2001).

These *visual effects* are often being seen as a very important factor for socio-political as well as local acceptance of wind energy (Breukers and Wolsink, 2007; Devine-Wright, 2005; Haggett, 2011; Jones and Eiser, 2010; Wolsink, 2007a). Wolsink states that "[…] visual evaluation of the impact of wind power on the values of the landscape is by far the dominant factor in explaining why some are opposed to wind power implementation and why others support it" (Wolsink, 2007a: 1194). Citing a study with residents of wind turbines in Sweden (n=351, Pedersen and Persson Waye, 2004) he comes to the conclusion that even noise is less important (Wolsink, 2007a). This last point can be questioned since Langer and colleagues report a somewhat different order. They apply a hypothetical choice experiment on a panel sample of 1.363 Germans aged 18 or above. The three attributes with the highest average relative importance values with respect to the acceptance of local wind energy projects are sound level at place of residence (first place), distance to place of residence (second place) and participation (third place). Visibility at place of residence is less important (Langer et al., 2017).



Visual effects are only relevant if the affected landscape is relevant to the people living there[vii]. In this context, place attachment is often an important impact factor for the acceptance of wind turbines (Bell et al., 2005; Jones and Eiser, 2010; Liebe and Dobers, 2019). *Place attachment* can be defined as "[…] positive emotional bonds between people and valued environments […]" (Devine-Wright, 2007: 7). Important to say, place attachment can have positive or negative effects on

local acceptance of wind energy depending on whether or not a wind farm fits to the meaning of the place (Devine-Wright, 2007; Di Nucci et al., 2020). Liebe and Dobers found that place attachment and protest intentions against renewable power plants (wind energy, solar energy, biomass) within a 10-kilometer radius of the respondent's place of residence are positively correlated (Liebe and Dobers, 2019: 253). Jones and Eiser compare attitudes towards wind power in the UK and proposed sites for wind turbines, respectively, of a sample from five cities near Sheffield which are all located within 1,5 km to these

proposed wind turbine (n=417, "target towns") with attitudes towards wind power and wind turbines, respectively, of a sample from five cities that are further away (n=392, "comparison towns"). They report for the target towns that the effect of considered site visibility on acceptance of local wind turbines was only apparent when the respondents held concerns about the landscape (Jones and Eiser, 2009; 2010: 3112ff.).

But the findings aren't always consistent. For example, Firestone and colleagues interviewed a large random sample of wind

turbine residents in the USA (n=1.705) by telephone, internet and mail to investigate the perception and evaluation of local wind energy projects (Firestone et al., 2018). Contrary to the European findings, they didn't reveal any effect of place attachment on the attitude towards local wind energy projects. The researchers also find "[…] that project appearance in general (its look) matters more than whether it fits the landscape" (Firestone et. al 2018: 379). Referring to value orientated approaches like the Cultural Theory (Douglas and Wildavsky, 1993; Wildavsky and Dake, 1998), different values imply

specific bias in perception and could serve as a possible explanation. Of course, this would question the transferability of national research results (Firestone et al., 2018, see also Aitken, 2010a). But this has to be tested by future research. All in all, high value of local places and perception of negative visual effects of wind turbines probably form a powerful source of resistance against such developments.

### 3.2 Proximity effects

It is clear that visual effects can only take part if the acceptance object can be seen (Bishop, 2002). This means that moving the object farer away could lead to higher acceptance ("out of sight, out of mind"). This rationale is linked to the so called "proximity hypothesis" which states that "[…] those living closest to a wind farm will have the most negative perceptions of it […]" (Harper et al., 2019: 956). Accordingly, wind energy projects should be more acceptable when they are realized offshore (meaning somewhere on the ocean) than onshore (somewhere in the country) or on the local level (somewhere in

my neighbourhood). The empirical evidence for this hypothesis is rather mixed (Devine-Wright, 2005; 2007; Di Nucci et al., 2020; Harper et al., 2019; Jones and Eiser, 2010; Reusswig et al., 2016; Swofford and Slattery, 2010; Wolsink, 2007a; b). For example, on the one hand, Jones and Eiser state that the acceptance of wind energy in the UK is in the target town group higher on a national level compared to the local level. In general, acceptance rises as distance increases. It is lowest on the





local level, higher for onshore wind and highest for offshore wind (Jones and Eiser, 2010). This finding was successfully
replicated by Sonnberger and Ruddat for Germany (Sonnberger and Ruddat, 2017) as well as Hübner and Hahn for three
regions in Germany (non-representative-sample, n=704, Hübner and Hahn, 2013). Swofford and Slattery also find evidence
in favour of the proximity hypothesis on the basis of a mail survey in the USA conducted in 2009. They use a random
sample of 200 residents of a wind farm with 75 wind turbines in Texas. The distance of respondents' home to the wind farm
was up to 20 km. They report "[…] an inverse relationship between proximity and positive attitudes, whereby acceptance of
wind energy decreases closer to the wind farm […]" (Swofford and Slattery, 2010: 2514). In the same way, Langer and
colleagues discover a relatively high relevance of distance to place of residence for the acceptance of wind energy projects
whereas "[…] respondents preferred larger distances between the wind turbines and their place of residence" (Langer et al.,
2017: 68).

On the other hand, Hoen and colleagues report positive effects of proximity of wind turbines on acceptance using a randomly
selected sample of residents near wind turbines in the USA (n=1.705, Hoen et al., 2019: 7). Wolsink interviewed 531
environmentalist (members of the Wadden Vereninging) and found no effect of distance on the attitude on the siting of wind
turbines in the Wadden region (Wolsink, 2007a: 1199). Hübner and Pohl summarise findings of four studies with residents
of wind turbines in different regions of Germany and Switzerland (212 < n < 467). Measuring distances on a metric scale
(for example from less than 600 m to more than 2000 m) they found no correlation between distance to the nearest wind
turbine and acceptance of wind energy in general as well a locally (Hübner and Pohl, 2015: 11).

Differences in results are not very surprising given the fact that the studies vary with respect to acceptance subject,
acceptance object and acceptance context.  Because social science research (especially in an international context) has
always to deal with some degree of cultural variation, this is just natural. On the other side, this highlights the great
importance of longitude research and cross-national studies. For example, Wolsink reports a U-shaped curve of wind energy
acceptance as a result of experimental studies conducted in the Netherlands (pre-post-test control group design, 333 < n <
680). He differentiates between three phases: before project planning, during the siting process and after the wind turbines
started running. Although general attitudes towards wind power as well as local acceptance of wind farms are positive on
average in all three phases, they are high in the first phase, relatively low in the second phase and high again in the third
phase (Wolsink, 1988; 1994; 2007a; b). This positive effect of direct experience with the risk source has also been found in
several other studies and countries (e.g. Ireland, Scotland and USA; Krohn and Damborg, 1999; Swofford and Slattery,
2010; Warren et al., 2005). It can probably be traced back to overexaggerated expectations about the negative environmental
impacts of the wind turbines (van der Horst and Toke, 2010; Warren et al., 2005). But it is certainly not an automatism.
Wolsink states that "[…] it is by no means a guarantee for improvement of attitudes after a facility has been constructed. The
effect can only be seen if the existing environmental impact is adequately dealt with, in the eyes of the local population"
(Wolsink, 2007a: 1199)[viii]. These research results help to explain at least in part the differing results with respect to the
proximity hypothesis: wind turbines near to residential areas can have a negative effect on acceptance in case of *proposed*



*sites* but a positive effect in case of *existing sites*. This differentiation between proposed and existing sites is also emphasized in the literature (van der Horst, 2007; Hoen et al., 2019; Swofford and Slattery, 2010; Warren et al. 2005).

The proximity hypothesis can also be linked to the famous but meanwhile outdated NIMBY ("Not In My Backyard")
phenomenon (e.g. Aitken, 2010b; Breukers and Wolsink, 2007; Devine-Wright, 2007; van der Horst, 2007; Jones and Eiser, 2010; Sauter and Watson, 2007). It means "[…] that people have positive attitudes towards something (wind power) until they are actually confronted with it, and that they then oppose it for selfish reasons" (Wolsink, 2007a: 1199). NIMBY is problematic for at least three reasons. First, it is certainly not the only explanation for resistance (alternatives are for example place attachment or a lack procedural fairness, Jones and Eiser, 2010; Wolsink 2007a). Second, it is a very simplistic form of
explanation (i.e. there are certainly more reasons for human behaviour than just selfishness, Bell et al., 2005; Devine-Wright, 2007; Wüstenhagen et al., 2007). Third, it is a one-sided negative label for respondents ("[…] it is never a compliment to call someone a NIMBY […]" Haggett, 2011: 504). Although the negative consequences of the NIMBY concept are clearly acknowledged here, there is one convincing argumentation by Bell and colleagues who connect NIMBY to Rational Choice Theory in order to explain the social gap:

"The Nimby explanation of the social gap is the only explanation that depends upon an individual gap between attitudes to wind power in general (unqualified positive) and attitudes to a particular development (negative) […] On the Nimby account, the individual gap is the gap between collective rationality (or concern for the public good) which people will express in opinion surveys when it costs them nothing and individual rationality (or self-interest) which will motivate their behaviour" (Bell et al., 2005: 465)

This means "collective rationality" refers to the general support of wind energy (i.e. socio-political acceptance) while "individual rationality" refers to local acceptance. This is in line with research findings referring to several distance measures of wind energy projects (local, onshore, offshore) instead of one overall measure of wind energy acceptance.

### 3.3    Trust

The role of trust for risk perception, risk management, (risk) acceptance and facility siting has been well researched in the
last decades (e.g. Butler et al., 2011; Earle and Cvetkovich, 1995; Johnson, 1999; Renn and Levine, 1991; Slovic, 1993; Wüstenhagen et al., 2007). For example, the moderating effects of trust on risk and benefit perceptions are well known. If trust in relevant actors (e.g. official agencies, scientists, environmentalists, industry) is high, benefit also tends to be rated high and risks low and vice versa. This in turn has effects on risk acceptance (Siegrist, 2000; 2001).

*Trust* can be defined as "[…] a feeling or belief that someone (or some institution) will act in your best interest" (Bellaby,
2010: 2615). But why should someone or some institution do that for me? Earle and Cvetkovic argue that trust (or social trust as they call it because it is socially constructed) is based on value similarity (Earle and Cvetkovich, 1995[ix]). People sharing common values can more easily trust each other. In the modern, complex world there are many new technologies and associated risks that no one and no institution can handle alone (Renn, 2008). This is one reason for the importance of trust. Additionally, a lot of people don't know much about these technologies. In such a situation of high complexity and



little knowledge, trust becomes even more important. It reduces complexity to a certain degree and creates possibilities for joint action. Of course, trusting someone or some institution is a risk by itself because expectations always can be disappointed. In this case, trust is lost. In fact, it is lost very easily and regaining it is (very) difficult (Huijts et al., 2007; Kasperson et al., 2003; Luhmann, 2014; Siegrist, 2001; Slovic, 1993).

Scholars regularly cite two central elements of trust: competence and care. *Competence* entails the technical knowledge and

capabilities to rationally manage risks. Empirical indicators can be for example education, qualification or perceived performance in risk management. *Care* refers to the perceived responsibility to manage risks in the right way which means acting on the basis of shared cultural values. Empirical indicators can be respect of the common good or honesty (Johnson, 1999; Huijts et al., 2007; Renn and Levine, 1991; Zwick and Renn, 2002).

The *concept of trust* has been also used in the context of renewable energies in general and specifically wind energy. For

example, Aitken found in a Scottish case study some hints for the effects of distrust on the perception of unfair processes in the siting of wind energy projects. He notes that "[…] initial suspicions that the developers[x] would not act in the community's best interests led individuals to view decision-making processes concerning the development to be unfair. From the earliest stages the community benefits package was perceived as representing a bribe […]" (Aitken, 2010b: 6074). Sonnberger and Ruddat deliver mixed evidence for the role of trust. On the one hand, a multiple regression analyses revealed

only two significant correlations (out of twelve possible ones). Trust in big energy companies and the acceptance of offshore wind farms correlates negatively and trust in big energy companies and the local acceptance of wind farms correlates positively (Sonnberger and Ruddat, 2017). On the other hand, a categorical principal component analysis with the same data showed the relevance of trust for risk perception of renewables. The analyses revealed risk-benefit/acceptance and trust/fairness as the two main latent dimensions underlying citizens' perception of the German energy system transition

(Sonnberger and Ruddat, 2018)[xi]. Jones and Eiser report effects of trust in the target town group of their study in Sheffield: "[…] the more target respondents trusted Sheffield City Council to act with due fairness and transparency when furthering their plans for wind development, the more likely they were to hold favourable attitudes towards development, and vice versa" (Jones and Eiser, 2009: 4609). Hall and colleagues conducted a qualitative case study on wind energy in Australia (guideline interviews, n=27). Municipality officials advised wind developers to be frank and open in communication

processes to build up trust and the wind developers used local multiplicators to generate trust (Hall et al., 2013). The relevance of local relationships and local foundation is also mentioned by Hübner and colleagues (Hübner et al., 2020) as well as Haggett (2011). All in all, trust seems to be a key element for acceptance: "[…] establishing trust between the wind developer and affected stakeholders throughout the life of the wind farm is a theme with significant impact on the resulting level of social acceptance […]" (Hall et al., 2013: 204).

**3.4      Risk and benefit perceptions**

The application of technologies always implicates benefits as well as risks (Fischhoff et al., 1981, Perlaviciute and Steg, 2014). There are no universally ideal options for the satisfaction of human needs like transportation, food, housing or energy



production. For example, nuclear power provides on the one hand seemingly endless energy without the carbon emissions of fossil fuels. On the other hand, society has to deal with the catastrophic potential of a worst-case scenario (accident in a

nuclear power plant) and the still unsolved disposal problem of heat generating radioactive waste. As a consequence, public risk perception can be characterized by ambivalence with a strong tendency to rejection of the technology (European Commission, 2010; Gamson and Modigliani, 1989; Slovic, 1993; Zwick and Renn, 2002). Some authors speak at least of a "reluctant acceptance" of nuclear energy as a means to fight climate change (Butler et al., 2011; Corner et al., 2011).

The list of possible *risks and benefits* of wind energy is long. Hilary S. Boudet gives a rather comprehensive overview.

Economic development, tax revenue, landowner and/or community compensation, reduced air pollution and carbon savings are examples for commonly cited benefits for utility-scale wind. Examples for commonly cited risks for utility-scale wind are ecosystem impacts, visual impacts, sound annoyance and health effects as well as impacts to property values, electricity rates, tourism and so on (Boudet, 2019). According to Bell and colleagues, the perception of these positive and negative aspects of wind energy can be related to what they call "qualified support" meaning people tend to accept wind energy not

per se and unconditionally but instead only if certain conditions (i.e. an acceptable risk-benefit-ratio) are met. This is another explanation for the social gap (Bell et al. 2005). Perlaviciute and Steg address collective as well as individual costs and benefits of energy applications: "People tend to ascribe high collective costs and low collective benefits to fossil fuels, including oil, coal, and gas, and to nuclear energy, whereas they tend to associate renewable energy sources with high collective benefits and low collective costs" (Perlaviciute and Steg, 2014: 363). This positive view of renewables (including

wind energy) is not present on the individual level though. Irrespective of that, the relationship between costs and benefits on the one side and acceptance on the other side is clear for both levels: Higher perceived costs correlate with lower acceptance and higher perceived benefits with higher acceptance (Perlaviciute and Steg, 2014: 363).

There is plenty of empirical evidence for the connection between the perception of risk/benefit and wind energy acceptance (e.g. Jones and Eiser, 2009; Walter and Gutscher, 2013; Sonnberger and Ruddat, 2017). Jones and Eiser use amongst others

benefits like general economic benefit, opportunity to invest and cheaper electricity. Risks are for example the spoiling of the landscape, the lowering of house prices and a general unwanted change. Results show that these benefit and risk perceptions correlate significantly with specific attitudes towards wind energy turbines (i.e. local acceptance, Jones and Eiser, 2009). Walter and Gutscher used an experimental setting in a rural community in Bavaria (Germany) to analyse the effects of different wind energy projects on the perception an evaluation of 350 respondents to a postal survey. Projects varied amongst

others with respect to the implementation/result of a citizens' vote and the existence of local benefits. They found a significant effect of regional benefit (e.g. a community fund) on the support of specific wind projects (i.e. local acceptance, Walter and Gutscher, 2013). The study of Sonnberger and Ruddat revealed significant correlations of perceived risks of wind energy (index containing spoiling of the landscape, noise and danger for birds) as well as perceived benefit (creation of new jobs) and the acceptance of wind energy (offshore, onshore and local, Sonnberger and Ruddat, 2017).



### 3.5 Fairness and participation

Social scientists have repeatedly emphasized a demand for participation in case of siting decisions (Allen, 1998; Rademacher et al., 2020; Renn 2004). Residents perceive possible negative impacts of infrastructure planning (e.g. roads, power plants, waste facilities) for human health and the environment in their neighbourhood. Because the risks are solely taken by the local population while the whole society benefits from the infrastructure, questions of *distributive fairness* arise (Bell et al., 2005; Hall et al., 2013). A similar argument is contained in the "Green on Green conflict" which means risking the local environment for the sake of the global environment (e.g. Devine-Wright, 2007; Wolsink, 2007a; Swafford and Slattery, 2010)[xii]. Haggett cites several studies documenting the gap between local risk and global benefit of offshore wind parks. Although all people on earth will benefit from successfully fighting climate change, the risks (e.g. environmental damage, negative effects on birds, fishes, fishing industry and tourism) are taken by the residents of the sites (Haggett, 2011).

Another part of the puzzle is *process fairness* meaning the appropriate participation of residents and other stakeholders in the decision-making process (Aitken, 2010b; Devine-Wright, 2007; Hall et al., 2013)[xiii]. Research has shown effects of both elements on the acceptance of new infrastructure (Firestone et al., 2018; Hoen et al., 2019; Wolsink 2007b). For example, Hoen and colleagues find statistically significant positive effects of perceived process fairness on the attitude towards a wind turbine (Hoen et al., 2019: 7). Sonnberger and Ruddat report significant positive correlations between distributive and process fairness on the one hand and the acceptance of wind energy onshore and wind farms in a distance of 500m from the respondent's home on the other hand (Sonnberger and Ruddat, 2017: 61). The relevance of distributive as well as process fairness with respect to the acceptance of wind turbines also shows up in the qualitative study of Hall and colleagues (Hall et al., 2013: 205).

*Participation* of residents and other stakeholders in siting decisions is seen as one possible way to come to commonly agreed solutions (Aitken, 2010b; Jones and Eiser, 2009; Klinke and Renn, 2002; Ruddat and Renn, 2012; Wolsink, 2007a). Although there is certainly no guarantee for success, "good participation" raises the chances to avoid or minimize conflict (Alcántara et al., 2016; Webler, 1995; Renn, 2004; Ruddat and Mayer, 2020; Schweizer-Ries et al., 2010). Like Breukers and Wolsink put it: "Participatory decision-making is unlikely to turn people who fundamentally oppose wind power into supporters. However, conditional supporters […] may accept a wind project when they have been given an opportunity to influence the design" (Breukers and Wolsink, 2007: 2738).

People can participate directly in the planning process or financially. Krohn and Damborg report empirical evidence for the positive effect of financial participation on acceptance (Krohn and Damborg, 1999). Pasqualetti also emphasizes the benefit for land owners in the USA through wind turbines on their property (Pasqualetti, 2001). In a mixed-method design using surveys, qualitative interviews, focus groups and workshops, Schweizer-Ries and colleagues examined the perception and evaluation of wind, solar and biomass energy in different German case studies and came to the conclusion that financial as well as planning participation can have positive effects on (local) acceptance (Schweizer-Ries et al., 2010). Hübner and colleagues arrive at the same conclusion (Hübner et al., 2020).



With respect to planning participation, Firestone et al. report a positive correlation between the perceived possibility of the community to influence the outcome of the siting process and the attitude towards the respective wind project (Firestone et al., 2018: 377). In the study of Langer and colleagues, participation was under the three attributes with the highest average relative importance values with respect to the acceptance of local wind energy projects (Langer et al., 2017: 68). Based on empirical evidence from several studies, Haggett asserts that "[…] opposition can be both because people perceive that they have no voice, or no power" and concludes that "the planning process for offshore projects should therefore ideally allow local people to have some say or even influence in the project […]" (Haggett, 2011: 507).

## 4        Discussion, conclusion and some open questions

The goal of sustainable development means to change the energy supply in Germany and elsewhere from the use of fossil fuels and nuclear energy to renewable energies like wind, solar and biomass. Wind energy is of major importance for the success of this transformation process because it can provide large amounts of relatively cost-efficient energy. This transition process is not an easy task. Technical challenges (e.g. distribution and storage of fluctuating energy) have to be handled and social aspects (e.g. the relation of lost and created jobs in the energy sector, new energy infrastructures in the vicinity of residential areas) have to be considered. This leads to the question of acceptance.

The theoretical concept of acceptance is complex as well as multidimensional. It encompasses attitudinal and behavioural elements and can be measured in many different ways. Additionally, acceptance object, subject and context, respectively, have to be distinguished. With respect to wind energy, this means differentiating between for example socio-political, market and community acceptance, respectively, on the object dimension (Wüstenhagen et al. 2007); the general public, stakeholders and residents on the subject dimension and countries with different cultural and/or institutional backgrounds on the context dimension. The definition used here stated that acceptance refers to a positive evaluation of an object (like wind energy or wind parks) by a social group (e. g. stakeholders, residents, the public) under certain circumstances (e. g. cultural or institutional context) that can have consequences for individual behaviour. This definition was the basis for the comparison of (quantitative) studies. In general, socio-political acceptance of wind energy is clearly positive in many countries of the European Union. This positive evaluation is even more apparent in comparison with conventional energy sources like coal, gas or nuclear power. Problems arise when looking at local acceptance of wind turbines or wind farms, respectively. The protest potential with respect to wind projects is clearly higher than the one for solar projects.

Social science research has identified several empirical drivers of socio-political and (even more important) local acceptance. On the one hand, there is empirical evidence that perceived individual as well as collective benefits (e. g. creation of new jobs), trust in relevant actors of the transition process, procedural and distributional fairness, good management of environmental impacts (e. g. visual effects, noise, fit with the landscape) as well as early and effective participation in planning processes and financial involvement can have positive effects on acceptance of wind energy in general as well as specific wind projects. On the other hand, perceived risks (e. g. spoiling of the landscape, danger for birds, noise), the





355 perception of missing procedural and distributional fairness, distrust in relevant actors of the transition process, bad management of environmental impacts as well as too late and/or ineffective participation in planning processes ("alibi participation") can have negative effects on acceptance of wind energy in general as well as specific wind projects.

Of course, this is a rather rough summary of the state of research. Chapter three has presented numerous quantitative as well as qualitative studies which demonstrate rich and diverse findings. They differ in accordance to research design as well as

360 place and date of research. Or in other words: They differ with respect to acceptance object, subject and context. On the one hand this accounts for the complexity and multidimensionality of acceptance and is therefore an advantage. Ideally, if findings can be replicated under varying situations, they stand on a more solid ground.

On the other hand, it raises questions concerning the comparability of research results[xiv] and is as such a disadvantage. Although quantitative studies using large (representative) samples and statistical analyses produce seemingly "hard"

365 evidence, they may differ with respect to sampling, wording and other relevant aspects. Because of that, it may seem difficult to compare different quantitative studies. Indeed, this is sometimes challenging and it should always be kept in mind that conceptualisations and measurement usually differ between quantitative studies at least to a certain degree (Aitken, 2010a). The case is even more complicated for qualitative studies. They include a quite different kind of sampling, data collection and analyses, respectively. It is indeed the individual context of the research situation that enables the researcher

370 to interpret the results. Comparability is only (if at all) possible on a very abstract level (e. g. the categories of content analyses). In every case, the reader should always have a closer look at the research design to be aware of the differences. This also highlights the importance of longitudinal as well as cross-cultural studies (e. g. Breukers and Wolsink, 2007; IASS 2020; Steentjes et al., 2017).

Another relevant question arises with respect to distributional fairness. Although empirical research has identified the fair

375 distribution of benefits and risks as an important factor of wind energy acceptance, it is still unclear what this exactly means. Who considers what distribution of benefits and risks under what kind of circumstances as fair? This would probably be a good starting point for more valuable cross-cultural research in this field.

Participation in planning processes for the siting of infrastructure (e. g. wind farms) is usually assumed to be case specific and context dependent (Aitken, 2010b; de Vente et al., 2016; Nanz and Fritsche, 2012). Because the site and the surrounding

380 environment as well as the cultural and social background vary, participation concepts and corresponding measurements have to vary, too. This is certainly true. But it is also very time demanding, costly and inefficient. The question arises if it is possible to find certain types of siting processes and to develop suitable types of participation concepts for them. The types of siting processes for wind farms could be structured by place (e. g. offshore vs. onshore), size (e. g. few wind turbines vs. many wind turbines) or social factors (e. g. rural vs. urban areas). These are just a few ideas from the existing research.

385 Future comprehensive studies including a mixed quantitative-qualitative design could of course reveal other important structural factors.

Finally, the mixed evidence concerning the proximity hypothesis leads to the question whether or not they can be integrated. One possibility linked to the empirical results concerning the U-shaped form of acceptance could be the differentiation be-



tween proposed and existing wind farms. But this would just be a starting point on the road to a more comprehensive
theoretical concept or framework. Examples for such integrative theoretical frameworks are the Elaboration Likelihood
Modell (ELM, Petty and Cacioppo, 1986; Petty and Wegener, 1999) or the Social Amplification of Risk Framework (SARF,
Kasperson et al., 1988; Kasperson et al., 2003; Renn et al., 1992). Additionally, it could be asked how proximity can be
operationalized in the right way since the relationships between the distance of wind farms and the local acceptance of
residents seem to vary with scales. If distances are measured on a metric scale, there is no relationship. If distances are
measured using ordinal scales (e. g. 500 m, 5 km, onshore, offshore), relationships show up as expected. It is probably not
just the physical distance alone that constitutes opposition or support but the *meaning* of the different distances for the
residents (i.e. social construction of distance, Devine-Wright 2005). What do they perceive to be their neighbourhood? How
big is it really? This may vary between social groups as well as between different cultures. Taken together, this review has
shown that despite a lot of valuable scientific work done until now, there are still some open questions left.

**Competing interests:** The author declares that he has no conflict of interest.

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
