# Peer review of "Public Acceptance of wind energy – Concepts, empirical drivers and some open questions"

_Wind Energy Science, 2021_

## Author Response (AR1)

| Reviewer Number | Reviewer comment | Author comment | Changes in article |
|---|---|---|---|
| 1 | This submission is a review article, and it claims to cover "central theoretical concepts as well as qualitative and quantitative empirical findings of social science research concerning the acceptance of wind energy in Germany and elsewhere". | This critic is about chapter 2 concerning theoretical concepts of acceptance. Actually, this chapter wasn't intended to tell something about the "true nature" of acceptance. It was simply intended to give a very short overview about known concepts that are used in the context I do research in the field. I tried to connect these concepts analytically, but according to revier#1 I failed doing so. For example, socio-political acceptance definitely encompasses more than just public attitudes towards a technology and is not an object. Institutional elements (see below) play an important role, too. Fortunately, these connections are not really necessary here and I will now simply list the concepts as an overview before my own definition used in the article (and omit the "theoretical", which has perhaps invoked too high expectations). | I have reorganized and partly rewritten chapter 2 to make my focus more clearer and avoid misunderstandings. See track changes for details. I have also added some central points in the discussion section (chapter 4). See also track changes for details. |
| 1 | Although the MS (manuscript) says the "concept of acceptance is complex as well as multidimensional" (l.338), its treatment in this review seems to be almost flattened to one dimension.In the conceptual treatment, essentially different concepts are treated as if they were similar, or otherwise used interchangeably (without notice, and with no explanation). | Indeed, I have a mainly sociopsychological perspective and my focus is on public (or individual) acceptance of wind energy, wind turbines and wind farms. There are many more perspectives as reviewer#1 rightly points out. But this would be on the one hand too much material for the article and on the other hand isn't really in my expertise. So, public acceptance is indeed only a part of the puzzle and I will certainly acknowledge that in the second version of the article. The concepts I briefly described fit to the empirical data used. Of course, this data is also limited. So, a suggestion could be to do more studies in a comprehensive framework consisting of institutional context (financial system, support programs, legislation etc.) and culture. This would be a promising approach for the future and I will add this to the discussion section. I certainly make no normative judgement about the role of public/individual acceptance on the successful realization of wind park projects that goes beyond the statement that it has some effect. I do not state how big this effect is (and I will revise the criticized sentence in line 30 to make it more suitable to that position). Certainly, there are many impact factors and public/individual acceptance is just one of them. My definition of acceptance actually implies two dimensions (attitudes and behaviour) and because of that, different forms of measurement. This is meant with the quote in line 338 ("The theoretical concept of acceptance is complex as well as multidimensional. It encompasses attitudinal and behavioural elements and can be measured in many different ways"). I absolutely agree with the reviewer that acceptance is dynamic. Actually, this dynamic was shown in my presentation on the WES 2021 with respect to the different concepts of acceptance (analyses, feelings, behaviour). Of course, the weighting of benefits and risk is a process as is the formation of feelings towards a new technology or the way to the decision about protest or support of a local project, respectively. But at the end, there stands a judgement and this judgement can be measured by survey techniques (with all the methodological advantages and disadvantages, of course). So, it is advisable to look at acceptance at different points in time and this is done in the article by referring to data from a longer time span. Important to say, longitudinal research is of high value with respect to the dynamic nature of acceptance. Because of that, the case of the U-shaped form of acceptance (see line 180ff.) is of special interest and mentioned in the discussion. By the way, at least in the German case, public acceptance of wind energy is relatively stable over the past years, so although acceptance can be dynamic this doesn't mean it actually has to be. It is an empirical question. | |
| 1 | As such, the review contains a lot of relevant observations that might be interesting for readers with other disciplinary backgrounds, e.g. for WES. So, it could be potentially valuable, but it is conceptually badly organized, generating substantial confusion. As a result, there is on the one hand only a very limited understanding of the problem, on the other hand the MS contains a broad claim to cover 'acceptance'. Social acceptance is not a static phenomenon (a perception or attitude), but a dynamic process; above all, it is social, not just public; and it covers various areas of wind energy deployment and application. | | |
| 1 | 1. In the intro, the key problematic issue of this review immediately comes to the fore. Most problems for further deployment of wind, as described in this review, concern structural and institutional system characteristics. Transforming those seems to be the key to wind energy, so the transformation of the energy system and adjacent domains (e.g., land use planning) should be considered. Nevertheless, it is it's worded completely in reverse: "wind energy if of major importance for the success of the energy system transformation". This reveals how this review currently lacks a clear conceptual approach to both the concept of acceptance and the object of the acceptance process. The energy system/power supply system is a social-technical system (see all literature on these transformations/transitions; e.g. Geels among many others), and its major required transformations are also socio-technical. The acceptance issues concerns all such socio-technical changes, that is social changes combined with technology changes. | | |
| 1 | 2. Within the same paragraph, the object of acceptance (wind energy) seems to change without notice: (line 30) "All in all, acceptance of wind energy projects seems to be of crucial importance." This raises three fundamental questions:
- Are the acceptance object 'wind energy' and 'wind energy projects' the same?
- If not, is there arelation, and if there is (yes, but a weak one) a relation, what is it? - And if we deal with acceptance processes of projects, do we cover the issue of acceptance of wind energy? Unfortunately, for the current state of the MS, the answer to all 3 questions obviously is 'no'. All literature is illustrating the huge difference between the two, which is much more than the assumed 'gap' that is mentioned; in fact acceptance of a RE source/technology is only weakly correlated to acceptance of projects in which RE is implemented. And this relation is complex, variable, and depending upon many different elements of acceptance of RE innovation. | To make it clear, in this article, acceptance refers to the objects of wind energy in general and especially to wind energy projects (wind parks, wind farms, wind turbines) and the subjects of the public or parts of it (e. g. individuals like residents of wind parks, stakeholders). These are elements used in the empirical data available to me at the point of writing the article. Maybe I made a mistake at the beginning of the final paragraph on chapter 2 since this could imply some sort of "true nature" of acceptance. This is certainly not the case. The definition proposed at the end of the chapter is just the one suitable in my view for the following analyses. Since a lot of studies build on independent and dependant variables using multivariate modelling approaches the focus on individual attitudes and (self-reported) behaviour in the definition of acceptance lies near. Because reviewer#2 also had some problems with the definition I now expanded it to encompass possibly all types of attitudes and non-attitude:
"Acceptance means a positive evaluation of a topic (like wind energy, wind turbines or wind parks) by individuals under certain circumstances (e. g. cultural or institutional context) that can have consequences for individual behaviour. Correspondingly, non-acceptance means a negative evaluation of a topic by individuals under certain circumstances that can have consequences for individual behaviour. If there is no clear positive or negative attitude towards the topic (e.g. ambivalence, non-attitude), we speak of tolerance that can have consequences for individual behaviour (but perhaps not so much as the endpoints of the continuum)." Although I do not investigate the cultural or institutional context in much detail (because it is not the focus of the article), it is of course present (at least in a theoretical perspective) and should be part of the definition. Additionally, the empirical studies come from many different countries and so the institutional and cultural context is indeed varying. | I have modified the definition of acceptance in chapter 2 as described here. |
| 1 | 3. Illustrating the difference indicated in (2), very relevant for this MS is, e.g., the entire section on trust. It is hardly about 'wind', but all the more about projects (trust in developers; confidence in or perceptions of fairness; both not are not attributes of 'wind'). The hidden relevance of the object is also apparent, for example, from a sudden, unexplained switch to the concept (is it really a concept?) "utility scale" wind (267) "Utility" is an indication of a kind of actor/developer (says little about 'scale'), and even then rather fuzzy (private commercial? or public/state). Presumably very different from an initiative rooted in the community (common, corporate, local, etc.) that could operate on the same scale, but immediately entails a different picture of risk/benefits (and of trust). As it stands, the essential question of owner/initiator/operator is tucked away in one paragraph (line 316 and following), but this characteristic of the project is probably just as important as the fact that the project is about wind (could also cet. par. could have been solar, geothermal, hydro etc.). | There seems to be a problem with the term "Utility-scale wind" in line 267. This is a quota from the Boudet-Article. It does not mean introducing a new concept or something like that. I understood it as representing great wind parks and their benefits and risks. And this is what chapter 3.4 is all about. But I will change it into "large wind farms" to make it more clearer. The question of ownership/initiator/operator is certainly of high relevance. I incorporated it in the chapter about trust as well as participation because there are strong empirical relationships (see the cited literature in lines 251f and 316ff) and also for brevity. I am not sure if it really needs an own chapter because there would be some redundancy and it would make the article longer (and by the way it is long enough in my view thinking of the reader). | I changed "utility-scale wind" into "large wind farms". I added another sentence in chapter 3.3 (line 315-317 in the track changes) to highlight the importance of ownership as well as supporting literature (just one sentence for brevity). |
| 1 | 4. Hence, beside the issue of acceptance of what (the acceptance object) the most prominent issue is the acceptance by whom? The MS introduces 2 'theories' that are entirely different, in fact are targeting totally different things. The 'theory' of Wustenh. et al is an only an elaboration of a concept, so in fact describing a domain of academic research and in fact inviting several theories for explaining different elements and processes in SA of RE innovation. Applied theories in SA (e.g., for example focusing on the land use dimension only: Busse and Siebert, LandUsePol, tab2) are so far mostly fundamentally single discipline and all focus on one specific layer of social acceptance. In real life, the acceptance is about a variety and complexity of systems (an example of that could be CPR theory on the exploitation and use of natural resources, and renewables are natural resources; see Ostrom); It is about poly-centricity and multi-level governance, self- governance, adaptive governance, and institutional flexibility. | Chapter 2 was simply intended to give a very short overview about known concepts that are used in the context I do research in the field. I tried to connect these concepts analytically, but according to revier#1 I failed doing so. Institutional elements (see below) play an important role, too. Fortunately, these connections are not really necessary here and I will now simply list the concepts as an overview before my own definition used in the article (and omit the "theoretical", which has perhaps invoked too high expectations). Indeed, I have a mainly sociopsychological perspective and my focus is on public (or individual) acceptance of wind energy, wind turbines and wind farms. There are many more perspectives as reviewer#1 rightly points out. But this would be on the one hand too much material for the article and on the other hand isn't really in my expertise. So, public acceptance is indeed only a part of the puzzle and I will certainly acknowledge that in the second version of the article. The concepts I briefly described fit to the empirical data used. Of course, this data is also limited. So, a suggestion could be to do more studies in a comprehensive framework consisting of institutional context (financial system, support programs, legislation etc.) and culture. This would be a promising approach for the future and I will add this to the discussion section. I certainly make no normative judgement about the role of public/individual acceptance on the successful realization of wind park projects that goes beyond the statement that it has some effect. I do not state how big this effect is (and I will revise the criticized sentence in line 30 to make it more suitable to that position). Certainly, there are many impact factors and public/individual acceptance is just one of them. | I have reorganized and partly rewritten chapter 2 to make my focus more clearer and avoid misunderstandings. See track changes for details. I have also added some central points in the discussion section (chapter 4). See also track changes for details. |
| 1 | 5. Without informing the reader, this MS remains limited to one specific layer, the acceptance by the public, suggesting this covers acceptance in society (a tragic misconception). Furthermore, it is confusing public acceptance of 'wind energy' (which is one element of socio-political acc.) and acceptance by local residents (which is mainly community acceptance, but also market acceptance in case of community initiatives and consumption of wnd generated power). The latter even without really structurally considering the layers of market and socio-political acceptance, that both are complex and full of elements that are acceptance objects in itself. | | |

| | | | |
|---|---|---|---|
| 1 | 6. Acceptance of wind energy is a set of processes in society, not simply a position taken by individuals. See W. et al., but also more recent further elaborations of the concept of SA of renewables' innovation. For example, Sovacool e.a., and most recently an overview by Gaede & Rowlands, with a critical discussion following (EnergyResSocSci 2018/9). Like in this MS, in many studies social acceptance is still translated immediately into acceptance of individuals, and more particularly individuals in the public. The authors seems to take the very limited view of Upham ea as a starting point, with heavy consequences. First, instead of acceptance as originally defined as a decision-making process, developing over time, with mutually affecting actors, and also on different levels or dimension (as elaborated by Wüstenhagen et al and the mentioned literature), it is by Upham reduced to a one-dimensional response by individuals only. Anything else is massaged away: the dynamics, the influence of organizations (private, commercial, authorities etc.). The most important acceptance objects are fully out of sight: decisions about the institutional conditions (mainly within the W's domain of socio-political acceptance; see (7) below). In Upham's narrow view these are 'acceptance context', which is a tragic mistake. Unfortunately, this MS also shifts very quickly, without further explication, to the discussion of public acceptance, rather than acceptance by the society. However, mostly other actors are paramount in the acceptance process, and their actions are definitely often not reflecting the positions taken by the public (some SA studies, by the way highlight the existence of many different publics!). E.g., in the example of acceptance of nuclear the authors explicitly the remark on acceptance to the public (262), as otherwise the remark is incorrect. Socio-political acceptance of nuclear is high in some countries (France recently announced re-regeneration and expansion; market acceptance too, as some energy companies are willing to invest [that is, if governments are covering the risks]), etc. | My definition implies two dimensions (attitudes and behaviour) and because of that, different forms of measurement. This is meant with the quote in line 338 ("The theoretical concept of acceptance is complex as well as multidimensional. It encompasses attitudinal and behavioural elements and can be measured in many different ways"). These elements are also part of the definitions of Upham et al. and Schweizer-Ries et al. and makes them at least two-dimensional in my view. In addition, Upham et al. incorporate "socio-technical system" as a possible acceptance object and in this way acknowledge the social dimension of technology. | I have modified the definition of acceptance in chapter 2 and deleted the short section about nuclear energy since it was only an illustrating example. |
| 1 | 7. For wind, however, the most prominent social acceptance (as defined in Wustenhagen e.a.) issues remain undiscussed. For example, in Germany currently the rapid development of wind deployment has imploded, due to the introduction of 'tenders' and 'auctions. We neither read anything about the acceptance of these key instruments (with heavy negative consequences for deployment), nor about the devastating consequences for other dimensions of acceptance (e.g., see Grashof 2019; 2021: "Wind projects developed by the local community were found to be most beneficial for local acceptance, but appear to face the highest challenges in auctions"). Comparable to for example a similar phenomenon in DK; see Kirkegaard et al. JEnvPolicyPlanning 2021. | Indeed, I have a mainly sociopsychological perspective and my focus is on public (or individual) acceptance of wind energy, wind turbines and wind farms. There are many more perspectives as reviewer#1 rightly points out. But this would be on the one hand too much material for the article and on the other hand isn't really in my expertise. So, public acceptance is indeed only a part of the puzzle and I will certainly acknowledge that in the second version of the article. | I have reorganized and partly rewritten chapter 2 to make my focus more clearer and avoid misunderstandings. See track changes for details. I have also added some central points in the discussion section (chapter 4). See also track changes for details. |
| 1 | 8. The remarkable shifts in the policy frame in D and DK are examples of a major acceptance issue, so an important acceptance object. It certainly is not simply 'context', but a set of key decisions in W's socio-political acceptance dimension. The following conclusion, therefore, in section 4 falls completely short: "...acceptance object, subject and context, respectively, have to be distinguished. With respect to wind energy, this means differentiating between for example socio-political, market and community acceptance, respectively, on the object dimension (Wüstenhagen et al. 2007)." However, the 3 dimensions of W. et al. are NOT objects. All objects relevant for wind deployment are present in all 3 dimensions; similarly the public, stakeholders and residents "on the subject dimension" operate in different roles and appearances, also in all 3 dimensions. Moreover, indeed countries may have different cultural background, but the decisions about the institutional frames of RE implementation (legislation, market structure, land-use regimes etc.) and the role of wind and RE projects, are the most prominent acceptance objects; it is not 'context'. | I agree with reviewer#1 that socio-political acceptance, market acceptance and community acceptance definitely encompass more than just public attitudes towards a technology and are not objects. So I do not make this connections any more in the article. | The named connections have been erased in the second version of the article (see track changes). |
| 1 | 9. The development of RE is many countries is often delayed, suffering from institutional barriers (lock-ins that are not addressed) and setbacks (institutional choices that create new obstacles). So, the remark that socio-political acceptance is high is very questionable, looking at the inertia in fundamental adaptions of policies, land-use and property regimes, and market-structures. So is the view that problems only 'arise' at the local level for projects, because the local problems, as described in the review, mainly arise as a result of the characterisics of projects that are initiated: problematic initiators and owners (distrust issue), the distribution of benefits issue, decision-making on land-use (e.g. the dominance of top-down siting, planning, and regulation), procedural justice issues, etc. Most of the elements in projects that are problematic for community acceptance, and for mrket acceptance, are determined by institutional frames. So they emerge from problematic socio-political and market acceptance dimensions. | Indeed, I have a mainly sociopsychological perspective and my focus is on public (or individual) acceptance of wind energy, wind turbines and wind farms. There are many more perspectives as reviewer#1 rightly points out. But this would be on the one hand too much material for the article and on the other hand isn't really in my expertise. So, public acceptance is indeed only a part of the puzzle and I will certainly acknowledge that in the second version of the article. | I have reorganized and partly rewritten chapter 2 to make my focus more clearer and avoid misunderstandings. See track changes for details. I have also added some central points in the discussion section (chapter 4). See also track changes for details. |
| 1 | (a) The main challenge for this review is to introduce conceptual clarity, necessary for a good judgement of all the factors that are discussed later. This is about a clear distinction between the object of 'wind energy' (as a source, or technology), and the object of decision-making about projects in which wind is only one of the main characteristics; beside that also developer, landscape (which is also an important attribute of a project, as alternative sites generate different landscape impact; see on the land use aspect for example two special issues of Landscape Research); other institutional frames (power supply; financial system, structure of governance etc.). However, this key issue may be repaired without completely reshuffling the article, but by disentangling the objects, and then implementing those conceptual picket posts consistently all over the paper. | | I have reorganized and partly rewritten chapter 2 to make my focus more clearer and avoid misunderstandings. See track changes for details. I have also added some central points in the discussion section (chapter 4). See also track changes for details. I do not speak of social acceptance in the article (and haven't done this before explicitly). |

| | | | |
|---|---|---|---|
| 1 | (B) The issues that we face with the conceptual fuzziness of the introduction in mind: why is acceptance in section 2 approached by a flat psychologic approach, by focusing on individual in the public? As the discussion in the paper now claims to be about acceptance, it is only reflecting the perceptions, attitudes and potential resistance of individuals in the public. Therefore, the wide claim of discussing 'acceptance' should be tuned down towards acceptance by individual members of the public only. Furthermore, it should also be clarified that this may be a relevant element of social acceptance, but only a small part of it. Public acceptance is not a proxy for social acceptance. Most actors and their roles in the process remain undiscussed, so this cannot be considered a review of social acceptance of wind. | | |
| 1 | Looking at visual impact, there is the phrase" that even noise is less important". Here is a reference to noise research by Eja Pedersen. Indeed, her research on noise annoyance is the most comprehensive so far. However, she did not reveal that noise is not important. Her investigations (done after the P & Persson-Waye paper) showed that noise annoyance is more strongly determined by visual assessment and (indirectly) project assessment than by objective sound pressure (e.g. Pedersen Larsman JEnvPsych 2008). Even more important: visual assessment/perception is not simply 'visibility', but a subjective assessment of the visual impact of the landscape change, the most important element in acceptance of land-use change for RE; also see some papers in LandscapeRes) This does not mean that noise is unimportant, (on the contrary, taking the noise issue serious is crucial in acceptance processes), only that on the individual level the subjective perception of annoyance in amplifying the effect of visual landscape assessment. | Concerning the remark on noise influence I will try to incorporate the findings of Eja Pedersen. I have a question here: If I understand you right, the visual assessment has some effect on noise annoyance. So, noise is only important if the object can be seen. Is that right? If so, I think it would be in accordance with the results of Pedersen and Persson Waye (2004). Concerning the importance of the subjective assessment of the visual impact of the landscape change, that's what I mean in line 126. | I added the study of Pedersen from 2008. |
| 1 | Aitken is 'she' (text 236). | | I changed this. |
| 2 | Are there really two different representative telephone surveys that both had N=2006, and both asked for the acceptance of solar farms, wind farms and high-tension power lines in 500 m distance to the respondents' home? You cite Schweizer-Ries and colleagues (2015) for one on l.74 and Sonnberger and Ruddat, 2016: 36 on l.95? This just seems unusually similar. | There is actually only one representative telephone survey with a sample size of 2006 that asked for the acceptance of solar farms, wind farms and high-tension power lines in 500 m distance to the respondents' home. The acceptance typology of Schweizer-Ries et al. was applied on this data using corresponding items. | No changes with respect to that point. |
| 2 | On l. 98-102 you conclude: "Altogether, acceptance means a positive evaluation of a topic (like wind energy or wind parks) by a social group (e. g. stakeholders, residents, the public) under certain circumstances". I don't agree that this definition is the most prevalent in the literature (nor should it be) and it's not consistent with what you said on l.40-71. As you point out, acceptance can also mean a lack of opinion or even grudgingly negative evaluation. For example, I don't like it if a police officer gives me a ticket, but I accept it as proper and legal. The idea is more complex than you are letting on. So I am perplexed as to why you are using this relatively narrow definition? | I have revised the acceptance definition to be more comprehensive: "Acceptance means a positive evaluation of a topic (like wind energy, wind turbines or wind parks) by individuals under certain circumstances (e. g. cultural or institutional context) that can have consequences for individual behaviour. Correspondingly, non-acceptance means a negative evaluation of a topic by individuals under certain circumstances that can have consequences for individual behaviour. If there is no clear positive or negative attitude towards the topic (e.g. ambivalence, non-attitude), we speak of tolerance that can have consequences for individual behaviour (but perhaps not so much as the endpoints of the continuum)." I hope this is better now. | I have modified the definition of acceptance in chapter 2 as described here. |
| 2 | L.106 you really evaluated the literature over several centuries? Must be a typo. You must mean decades. | Of course, it must be "decades" and not "centuries". That was a translation error. Thank you for the hint. | I changed "centuries" into "decades". |
| 2 | 112 it's obvious that wind turbines can have negative and positive impacts. What would be interesting is if you would tease out what makes some positive and others negative. | The reasons for the different positive or negative perceptions of the visual effects of wind turbines (line 112) are not very much detailed in the literature as far as I know it. Here are some examples:
"Whether wind turbines spoil or enrich the scenery, is a matter of taste" (Krohn and Dambourg 1999: 956).
"Existing empirical studies have indicated public support for turbines that are painted neutral colours and merge with the landscape. [...] In an extensive study of seven UK wind farm locations with 1286 respondents, they noted that 62% of respondents agreed that wind turbines symbolized 'a sign of progress', whereas 15% agreed that they symbolized a 'harking back to the past' and 16% agreed that turbines represented a combination of bot" (Devine-Wright 2005: 128f).
"Perhaps the issue revolves not around seeing a turbine per se, but rather around more aesthetic and socially constructed phenomena that account for how individuals evaluate a turbine's appearance and fit within the landscape. Based on our regression analysis, attitudes move from negative to positive as respondent attitudes move from not liking the look of the turbines and thinking they fit badly within the landscape to liking the look and believing they fit well" (Hoen et al. 2019: 9)
"The visual impact of a wind energy landscape is indeed important, but this impact will fluctuate greatly across unique locations and societies. Levels of environmental concern will surely differ by location and will depend greatly on local context and place attachment" (Swofford and Slattery 2010: 2514).
But I believe that the information about the possibility of negative as well as positive perception of the visual effects is indeed of some value because it highlights the variability of evaluations. | No changes with respect to that point. |
| 2 | l.113 a conclusion from 2001 that European wind farms are less ugly than American ones seems out of date and a gross overgeneralization that probably does not hold true today and if it does, you would need to tease out the subjective dimensions, not just offer up the opinion of Pasqualetti. There has been much more work done into how people perceive wind turbines than you reviewed here. | Thank you very much for the hint concerning the Pasqualetti paper. Indeed, it is a more qualitative description of the situation back in 2001 and explicitly marked as an essay. In that sense it may not be directly comparable to the other studies used in the review. But I found it to be very enriching especially because of the detailed descriptions in it (and he gives good reasons for his opinion). In addition, the results do not contradict other research results but complement them. And yes, the paper is not up to date but I think it is useful in a review process to incorporate data from many points in time, especially to see if there is some variation associated with the time dimension. For example, articles from 1999 to 2019 mention the different positive or negative perceptions of the visual effects of wind turbines (see citations above), so the finding can be seen as very robust. But I will make a notice to the subjective dimension of the Pasqualetti paper and see if I find some other, more up to date work concerning the perceptions of wind farms in Europe and the US. I would be thankful for hints with respect to literature from this special field. | Since I didn't find a more up to date paper that deals especially with this topic, I simply added an endnote with reference to Pasqualetti. |
| 2 | l.119 Wolsink did indeed say that, but there are lots of cases where economics was the dominant factor shaping support. Also, be careful about drawing generalities from Wolsink's article 15 years ago. Things have changed a lot, namely the size of turbines but also climate awareness, cost of wind electricity, etc. | I find it difficult to judge about the relative importance about the main factors discussed in my paper. That's why I list them equally in the discussion section. Wolsink makes a clear judgement that may be right (and his work is very prominent and often cited, so the finding may have stood the test of time). Other researchers point to other factors (e.g. benefits or financial participation that fit to the economic dimension you mentioned). Taking into consideration that wind turbines become bigger and move closer to residential areas, the visual impact may become even more important. | No changes with respect to that point. |
| 2 | l.144 I don't agree that cultural theory is not transferable across countries. Certainly, arguments made about political culture would be less transferable, but Mary Douglas's work has been noted to be quite cross national. See: Mamadouh, V. (1999). Grid-group cultural theory: an introduction. GeoJournal, 47(3), 395-409. | Cultural theory is definitely transferable across countries. This is maybe the main purpose of it: Comparing different cultural biases in different countries and the consequences for risk perception. I didn't mean anything else with mentioning it at that point of the article. But if there are clear different biases, the national research results are not transferable or at least not that easy transferable. I hope I could clarify this possible misunderstanding. | No changes with respect to that point. |
| 2 | l.154 Yes "offshore" but the point you are making, I believe, is that they need to be offshore far enough that they cannot be seen from anywhere on land, including a 37-story condominium. | Yes, offshore in the sense "somewhere on the ocean" means not to be seen or at least only seen as a very small, non-disturbing part of the horizon. I will add this. | I added a refinement to "offshore wind power". |
| 2 | 200-210 It's good that you mention NIMBY, but I feel that the topic is worthy of a great deal more consideration. It is a major argument made about acceptance. I feel it was giving too short consideration in this article. There is an extensive literature on NIMBY. Can you reference the key articles as it applies to wind? | Thank you for the remark about on NIMBY. Actually, I was wondering to mention it at all because there has been so much discussion about it and readers could say "Oh my god, not again, please". But I decided to incorporate the main arguments and counter-arguments as short as I could because it is still important. This importance lies in my view especially in the argument of Bell at al. For reasons of brevity, I would prefer to let it stand as it is. | No changes with respect to that point. |

| | | | |
|---|---|---|---|
| 2 | 235 You do a good job of explaining some of the nuances around the definition of trust. As you get into the empirical literature, it would be important to know how researchers have operationalized the term – in other words, which definitions are they using? | Definitions or concepts of trust within the referenced literature:
„Throughout its development, social trust was based on similarity of cultural values, and this was communicated within cultural groups by narratives constructed by community leaders. Social trust was socially based. [...] Social trust, in our interpretation, can be based on any values-whatever happens to be salient to a person at a certain time, in a given context." (Earle and Cvetkovich 1995: 19 / 105).
Butler et al.: No own definition of trust.
„Three types of factors that might affect trust – competence, care and consensual values – are proposed. Alliteration aside, these encompass the factors hypothesized in the literature. The first two are means to the ends (e.g. health and safety) that principals (e.g., citizens) presumably want agents (e.g. corporate or government officials) to achieve on their behalf, and for which trust in the agent might be warranted or needed. Consensual values can be either means or ends: means to the extent that (for example) someone with egalitarian values believes that egalitarian groups are most able to achieve health and safety; ends to the extent that equality is desirable regardless of health and safety trends." (Johnson 1999: 326)
„Trust in communication refers to the generalized expectancy that a message received is true and reliable and that the communicator demonstrates competence and honesty by conveying accurate, objective, and complete information." (Renn and Levine 1991: 179, accentuation in original)
Slovic 1993: No own definition of trust.
Wüstenhagen et al. 2007: No own definition of trust.
„Soziales Vertrauen wird dabei als Bereitschaft verstanden, sich auf andere zu verlassen und dabei das Risiko einer Enttäuschung in Kauf zu nehmen. In diesem Sinne ist Vertrauen immer »grundlos. [...] Vertrauen kann als Persönlichkeitsmerkmal betrachtet werden [...] Gewisse Personen zeigen eine stärkere Neigung, Vertrauen zu schenken als andere Personen. [...] Interpersonelles Vertrauen basiert auf direkten Interaktionen. Der Interaktionspartner kann beobachtet werden; Feedback ist also möglich. Dieses Merkmal fehlt beim sozialen Vertrauen."." (Siegrist 2001: 22 / 28 / 30, please let me know if there is a transition necessary)
„These results are not contrary to our perspective that trust should be viewed as a unidimensional construct ranging from trust to distrust." (Siegrist 2000: 196)
„Social trust is acquired actively in a repeating mutual process. In contrast to confidence, it is based on continued experience distinguished by certain qualities, e.g. credibility, honesty, reliability, a feeling of responsibility etc. [...] institutional trust is based on the perception and evaluation of specific performance: social institutions - industry, politics and authorities, the media, science and experts, but also environmental and consumer agencies - fulfill specific functions respectively, where the origin, research and communication, regulation and control of risks are concerned. According to his opinion, trust is not based on ›vague‹ faith, but, more robustly, on experienced performance, based upon which trust is granted or withdrawn." (Zwick and Renn 2002: 45 / 46)
I hope this helps for a better understanding. For reasons of brevity I couldn't incorporate them into the chapter. Because of that, the short definition of Bellaby was used. Of course, a | No changes with respect to that point. |
| 2 | 240 You write: "Trust in big energy companies and the acceptance of offshore wind farms correlates negatively and trust in big energy companies and the local acceptance of wind farms correlates positively (Sonnberger and Ruddat, 2017). " But this seems counter-intuitive and I wonder if it's a German result? Could it be instead that people's distrust of large energy companies is really linked to their perception of those companies' willingness to support Energiewende? We are talking about the Big 4, right? | One possible explanation for the contradicting results with respect to the big energy companies (in fact the big four in Germany, E.ON, RWE, EnBW and Vattenfall) could be the perception that these companies don't really support the energy transition. Actually, we had this result in focus groups (see Ruddat, M. / Sonnberger, M. (2015): Wie die Bürgerinnen und Bürger ihre Rolle bei der Energiewende sehen. In: et – Energiewirtschaftliche Tagesfragen, 1/2 / 2015, S. 121-125). I will add this in an endnote. | I added an endnote for explanation purposes. |
| 2 | 252-255 Your conclusion is underwhelming. Overall, trust matters. Yes, we know that. But how is trust built or lost in the wind industry? Is it different than in other technologies? Or, if the data are not there to allow you to conclude something interesting such as that, maybe it would be more useful to point out what kinds of studies are needed so that we get at the right questions? | As far as I know there are no differences between the creation or function of trust in the domain of wind energy compared to other technologies simply because trust is a general concept. The factors influencing the building of trust (for example showing or proving competence, objectivity in communication, see Renn and Levine 1991) can be used in any domain. But there are certain components in the chapter about fairness and participation that can have special impacts on trust. I will try to make these connections a little bit more clearer. | I added a connection between participation and trust in the fairness and participation chapter. |
| 2 | 269-271 This is an interesting finding – that acceptance is linked to meeting certain risk-benefit conditions. If so, this would mean a risk-based approach could be productive at resolving siting disputes. However, I imagine that qualitative dimensions of risk matter as well, and not simply a hard and fast number as produced by a technocratic approach to risk? What does the wind literature say about risk-benefit trade-offs and public acceptance? | Concerning a risk-based approach again I find it difficult to judge about the relative importance about the main factors discussed in my paper. I see them all in all as being equally important. And yes, of course, there are qualitative risk characteristics that are important for the perception and evaluation of risks (see the work in the tradition of the psychometric paradigm). But these incorporate personal and societal risks and benefits, respectively, as well. The question is how big have the benefits to be to outweigh the risks? And does benefit always mean monetary benefit or something else? There are some connections to the point in the discussion when it comes to distributional fairness (line 374ff.). So, I have that in mind. | No changes with respect to that point. |
| 2 | 309 Lots of people agree that participation can produce consent, but what does the wind literature add to this? Are there data about wind specifically (besides the Firestone 2018 piece)? Could you unpack that? It would be nice to learn what they found out. | The studies cited in the paragraph about participation are mainly with respect to wind energy (from line 313 up to line 329 at least). The results reported here are the things they have to add to the participation point. Of course, it is a rather rough overview, but for reasons of brevity I had to make a selection. | No changes with respect to that point. |
| 2 | l.316-323 this idea of financial reward and acceptance is important. I would really like to see the literature on this reviewed more thoroughly. For instance, what kind of financial participation did Schweizer-Ries identify? | Schweizer-Ries and colleagues report results of three focus groups conducted with citizens, planers investors as well as administrators and politicians in Germany in 2009. The question was how citizens could best participate financially in local as well as regional renewable energy projects (wind, solar, biomass and water) and what effects this participation has for acceptance. Results showed that trust in the developer, the approval of the project by the administration before advertising the possibility of financial contribution, a robust calculation of the project, regularly information about the returns and the setting of a minimum and a maximum value, respectively, for the individual contribution are some of the success factors. Although lower minimum values can lead to a broader participation, this may be a problem for the robust calculation of the project. Special projects for small and big investors, respectively, seem to be advisable. One example for financial participation formats are regional customer funds initiated by the regional energy supplier. These funds are exclusively for customers and guarantee a return without a risk for the customers. The saving deposit normally lies between 500 Euro and 10.000 Euro for five years enabling nearly everyone to participate in the project (Schweizer-Ries et al. 2010: 97ff.).
Krohn and Dambourg write with respect to financial participation: "In Denmark there is a tradition for wind co-operatives, where a group of people share a wind power plant. In that respect Sydthy municipality is quite unique with 58 pet. of the households having one or more shares in a cooperatively owned wind turbine. Regarding the general attitude towards wind turbines, the picture is clear. People who own shares in a turbine are significantly more positive about wind power than people having no economic interest in the subject. Members of wind co-operatives are more willing to accept that their neighbour erect a turbine" (Krohn and Dambourg 1999: 956).
Hübner et al. refer to the possibility of citizens to buy shares on local energy projects (e.g. wind parks). This possibility should be communicated early and transparently. Like Schweizer-Ries and colleagues they argue that lower minimum values for the individual contribution are better for a broader participation. Collectives allow for equal participation of every member irrespective of the individual contribution. This is a gain in fairness and can have positive effects on acceptance of local energy projects (Hübner et al. 2020: 27f.). | No changes with respect to that point. |
| 2 | 390 I was interested in what you meant by saying ELM or SARF are possible integrative theoretical frameworks. Where were you going with that? ELM is a model that explains how people can be persuaded to adopt certain attitudes or behavior, but SARF explains how risks are amplified through societal processes– although it's not a theory, as we've been told a million times. But how do you think these frameworks could be used to help answer the pivotal questions about wind? I don't see ELM and SARF as interchangeable at all, but perhaps that is not your claim? | ELM and SARF are simply two examples for models or frameworks that try to incorporate different research findings (ELM) or approaches (SARF). The question is if it is possible to develop a corresponding model for the acceptance of wind energy. | No changes with respect to that point. |
| 2 | Given all this literature that you have reviewed, and your findings that community acceptance matters, I would be most interested if you would entertain the question: Why do we continue to spend money on characterizing physical and biological impacts when so little is spent on working to build trust, recognition of benefits, how to do public participation well, the options for financial engagement, etc.? This is what frustrates me. We know wind turbines are not being sited for lack of public consent, but the answer is, instead, to do more research about impacts to bats and birds. I wonder if, after reviewing the literature, you agree with this position? | I totally agree with you stating a need for more social science research in the field of acceptance of wind energy since the solutions for a lot of research problems have not yet been found. But I am not sure if a review article is the right place for such a normative statement. Additionally, I myself am a social scientist and would perhaps profit from such funding, so I am of course biased. But maybe the editors could give a short information about whether or not such a statement they see as appropriate. | No changes with respect to that point. |

| # | Comment | Response | Changes |
|---|---------|----------|---------|
| 2 | This is a nice review of a lot of literature – mostly European, which is fine, but the real question is: How can we build popular and local support for more wind installations? What did you learn after reading all those articles? Can you summarize the best plan for action? Of, if the knowledge is not yet mature, can you recommend further research to do? | I think, my view on central open questions is being described in the discussion section already. For example, the development of suitable types of participation concepts for certain siting processes would be a great advancement in my view since it would help to give more structure and reliability in the social aspect of siting processes and probably help in gaining local support. So, this is one example for further research. | No changes with respect to that point. |
| 2 | Final point: I also would say that offshore wind is a totally different animal than onshore wind. Almost all the literature you reviewed is for onshore wind. That's okay since that's what most of the literature is, but I think you should note this more explicitly, add a caveat that says perhaps none of this applies to offshore wind, and point out the need for more work into offshore wind. | You are definitely right with the dominance of onshore wind in the literature. Like my review shows, there is some research done with offshore wind, so some of the results may apply to this, too (e.g. in case of the proximity hypothesis). But, yes, I will point to this discrepancy in the discussion section. | I added a new paragraph in the discussion section dealing with that point. |
| 3 | The paper starts with (see the abstract: "Problems are (…) and local protests") and endorses throughout an outdated – by scientific research standards – perspective to individuals' and communities' responses, specifically opposition, to wind farms as a/the problem. This is already and in itself clearly a significant limitation of this paper given that it aims to be an updated review of the literature in the area of the social acceptance of wind energy. In fact, the author not even refers to, let alone uses, some of the frameworks or approaches that have been more recently used in this area of research, such as those of energy and socio-environmental justice (e.g., Levenda et al., 2021, ERSS). In a related way, it is difficult to understand exactly why is this review innovative or even needed, especially given that recently there have been both thorough and insightful reviews of this field, starting with Wüstenhagen et al 2007, and going through Petrova, 2013, Ellis & Ferraro, 2016, Wolsink, 2018, Levenda et al, 2021,… To this regard, the author says by the end of the abstract that "Although there has been already a lot of valuable scientific work done, there are still some open questions left", but these open questions should be the key focus of the abstract and of the rest of the paper, given that the review of these relevant factors has been presented and discussed many times before ( and again, here, the review of these factors and frameworks is not even conducted in an updated way) | Maybe it would be good to explain how this article came into being since you are questioning the necessity of it. I was asked to present an overview about the acceptance of wind energy at the WESC 2021. I did it and based the overview on my scientific knowledge about the topic at that time. There were no complaints by the attendees (some of them with considerable experience in social science research in this field), about the (narrow) sociopsychological perspective or the main empirical drivers of acceptance. Since I assume that readers of WES and attendees of the WESC session could have at least some overlaps there was no reason for me to change a lot for the article (important to mention that presenters were given the opportunity to publish a written version of the presentation in WES). Of course, it has some gaps like every review article. Maybe, my one can fill some of the gaps in the reviews you mentioned. In addition, my open questions from the discussion section are still not addressed in the literature (as far as I know it) and can be seen as a contribution to the ongoing scientific research process in this field. With respect to the perspective and scope of the article (especially with respect to social acceptance), see also my comments on the text of reviewer#1. | No changes with respect to that point. |
| 3 | Additionally, there are also some conceptual incorrections in relation to the use of key concepts in this literature. For instance in the abstract, the author refers to perceived distributional fairness and to risk/benefits perceptions – but how are these different? The same applies to procedural fairness vs. participation – how are these different? | Distributional fairness contains a social element since the question is how risks and benefits are distributed between different groups in society (e.g. residents, stakeholders). Risks and benefits are simply lists of (more or less probable) negative and positive effects, respectively. Fairness and participation have indeed some connections and because of that I discuss them in the same chapter. | No changes with respect to that point. |
| 3 | The paper starts with a highly normative statement - The further development of wind energy is of major importance for the success of the energy system transformation in Germany and elsewhere - but without any data or arguments for backing it up. This should be addressed. | You are absolutely right that the statement "The further development of wind energy is of major importance for the success of the energy system transformation in Germany and elsewhere" is in a sense normative and needs some explanation. What is meant is that as far as I know it, wind energy is "technically" necessary for the success of the energy system transformation in Germany and elsewhere since it is more technically advanced than most other renewable technologies, is most economically profitable (so relatively cheap) and can be easily exploited. I forgot the relevant literature here and I apologize for that. I just wanted to be very brief at this point. The statement is supported for example by Cohen et al. 2014 (p. 4), Devine-Wright 2005 (p. 125), Ellis and Ferraro 2016 (p. 2 and 6) and Hagett 2011 (p. 503). Especially Ellis and Ferraro summarize it in a pretty good way: "Europe has some of the best wind resources in the world, providing a relatively cheap and exploitable renewable resource that has been core to many Member States' strategies for climate change and energy transition" (Ellis and Ferraro 2016: 2). The "problem of resistance" with effects like local conflicts, delay of implementation or abandonment of projects is also mentioned in the relevant literature many times (e.g. Breukers and Wolsink 2007, Cohen et al. 2014, Haggett 2011, Harper at al. 2019). I do not mark opposition to be deviant behaviour. This is why I also define non-acceptance. But if one takes the normative perspective of reaching a sustainable energy system, renewables are a main part of it. This leads to the question: "What are the factors that have an impact on local acceptance?" And this is the focus of my article (or at least one of them since I also deal with public acceptance). | I added a short explanation concerning the importance of wind energy in chapter 1 as well as supplementing literature. |
| 3 | Section 2 starts with: The iridescent concept of (risk) acceptance – for sure it is not the concept of risk acceptance that has been the focus of the extensive research on social acceptance that the author refers to. What follows is also then incorrect because what the author is referring to is to risk perception and not to the (social) acceptance of renewable energy infrastructures as a concept. Risk perception might affect acceptance but they are different concepts. | Some of the definitions cited use the term "risk acceptance" and some "acceptance" and this is why I set the "risk" here in brackets. Of course, the description is rather brief and the relationship between risk perception and risk acceptance could be presented in more detail (if I only had enough room for that) but it would not give much more interesting insights in the topic. Additionally, in the psychometric paradigm, risk perception (e.g. perceived catastrophic potential, perceived social hazard) is sometimes hard to seperate from risk acceptance. | No changes with respect to that point. |
| 3 | At a certain point the author says "under certain circumstances (e. g. cultural or institutional context) that can have consequences for individual behavior" – circumstances are not only cultural or institutional, but also or even mainly local and contextual, meaning that the way in which renewable energy infrastructures affect particular places, livelihoods, ecosystems… are contextual and to be understood at the community/place level, and not a the individual level only or eveb mostly. In fact, social acceptance is a very complex, multidimensional and dynamic phenomenon which is something that really is not integrated and accounted for in this review – this review presents acceptance in a simplistic and individualistic way (even if still in an unclear way regarding what could explain acceptance). | Cultural and institutional context are only two examples for circumstances. The context of the locality is surely another example and I can add this. And yes, like reviewer#1 already correctly remarked, social acceptance is complex, multidimensional and dynamic and dependent on myriads of variables (financial system, support programs, legislation, culture, local context et.). But I do not see a single study that encompasses all or at least many of these variables. There are serious problems of measurement as well as conceptualization (e.g. integration of qualitative and quantitative data, setting of adequate boundaries for the research area). This seems to be a big challenge for social science research that I cannot adequately address this in my article. This is why I concentrate on a mainly sociopsychological perspective and my focus is on public (or individual) acceptance of wind energy, wind turbines and wind farms. There are many more perspectives as reviewer#1 had already rightly pointed out. But this would be on the one hand too much material for the article and on the other hand isn't really in my expertise. So, public acceptance is indeed only a part of the puzzle and I will certainly acknowledge that in the second version of the article. | I have reorganized and partly rewritten chapter 2 to make my focus more clearer and avoid misunderstandings. See track changes for details. |
| 3 | Methodologically, the type of review conducted and the results taken from it is also highly problematic. The author concludes that: In general, socio-political acceptance of wind energy is clearly positive in many countries of the European Union. This positive evaluation is even more apparent in comparison with conventional energy sources like coal, gas or nuclear power. Problems arise when looking at local acceptance of wind turbines or wind farms, respectively. The protest potential with respect to wind projects is clearly higher than the one for solar projects. But a systematic and comprehensive review was not conducted nor statistics and results analyzed in a systematic way to arrive to such conclusions. Additionally, research highlighting the problems and limitations of much of the research on the social acceptance of renewable energy infrastructures is not here properly acknowledged and taken into account (e.g., Ellis et al., 2007; Walker, 2009; Aitken, 2010; Fast, 2013; Gaede & Rowlands, 2018: Wolsink, 2018, 2019,…) , thus delivering an extremely partial and incomplete – sometimes even incorrect – review of the literature in this area. | The summary cited ("In general, […]") is based on the last part of chapter 2. I cite several empirical studies to come to that conclusion (Devine-Wright, 2005; 2007; European Commission 2011, FA Wind, 2020; Krohn and Damborg, 1999; Liebe and Dobers 2019, Ruddat and Sonnberger 2019, Steentjes et al., 2017). Of course, this research (like every research) has limitations but these are not in the focus of my article because it is not a methodological one. | No changes with respect to that point. |